# Irregular Scene Text Detection Based on a Graph Convolutional Network

**DOI:** 10.3390/s23031070

**Published:** 2023-01-17

**Authors:** Shiyu Zhang, Caiying Zhou, Yonggang Li, Xianchao Zhang, Lihua Ye, Yuanwang Wei

**Affiliations:** 1College of Science, Jiangxi University of Science and Technology, Ganzhou 341000, China; 2College of Information Science and Engineering, Jiaxing University, Jiaxing 314001, China; 3Key Laboratory of Medical Electronics and Digital Health of Zhejiang Province, Jiaxing University, Jiaxing 314001, China

**Keywords:** text detection, scene image, irregular, relation inference, GCN

## Abstract

Detecting irregular or arbitrary shape text in natural scene images is a challenging task that has recently attracted considerable attention from research communities. However, limited by the CNN receptive field, these methods cannot directly capture relations between distant component regions by local convolutional operators. In this paper, we propose a novel method that can effectively and robustly detect irregular text in natural scene images. First, we employ a fully convolutional network architecture based on VGG16_BN to generate text components via the estimated character center points, which can ensure a high text component detection recall rate and fewer noncharacter text components. Second, text line grouping is treated as a problem of inferring the adjacency relations of text components with a graph convolution network (GCN). Finally, to evaluate our algorithm, we compare it with other existing algorithms by performing experiments on three public datasets: ICDAR2013, CTW-1500 and MSRA-TD500. The results show that the proposed method handles irregular scene text well and that it achieves promising results on these three public datasets.

## 1. Introduction

In recent years, with the popularity of smartphones and intelligent terminals, there has been a growing demand for the extraction of text information from scene images (such as those from intelligent transportation systems [1], geographic information systems [2], and automatic scene understanding [3]). In response, scene image text detection has attracted the attention of many researchers, and many text detection algorithms have been proposed, such as CTPN [4], FCEnet [5], and Quadbox [6]. However, the detection of irregular scene text is still an unsolved problem.

As methods based on connected regions are not affected by the lengths and shapes of text lines, they can achieve better performance in irregular scene text detection cases, as shown for CTPN [4] and TextSnake [7]. This kind of method usually employs convolutional neural network (CNN)-based approaches to predict the adjacency relations between text components. However, scene image texts are unfixed or irregular in terms of their positions, sizes, and orientations, and their underlying structures are non-Euclidean structures. When using a CNN to process such data, they need to be artificially converted to Euclidean-structured data, which can reduce the final performance, meaning these methods cannot directly capture relations between distant component regions due to the limitation of local convolutional operators [8]. On the other hand, graph convolutional networks have obvious advantages in processing non-Euclidean structured data. DRRG [8] detects irregular text via a graph convolutional network. This method generates components in each text region and then uses these text components to build a connected graph, where each node in the connected graph represents a text component. Then, a graph convolutional network is used to infer the adjacency relations between the nodes in the connected graph to generate text lines. However, this method produces a large number of noncharacter text components during the generation process, which affects the detection speed and performance of the approach. Inspired by DRRG [8], we propose an irregular scene text detection method based on a graph convolutional network. The main flowchart of our method is shown in Figure 1. During the text component extraction stage, to reduce the number of noncharacter text components, the estimated character center point is utilized after detecting the text regions with a fully convolutional network containing VGG16_BN as the backbone. In the text line grouping stage, we consider the process of grouping irregular text lines as a problem of inferring the adjacency relations of text components with a graph convolution network. Our paper makes two primary contributions.

(1) Different from the literature [8], we combine the text regions created by a fully convolutional network architecture based on VGG16_BN and the estimated character center points to generate text components, which can obviously reduce the number of noncharacter text components and help to improve the performance of text line grouping.

(2) We argue that the text line grouping problem can be posed as distinguishing edges that are adjacent and nonadjacent to an undirected graph. A relation inference network based on a graph convolutional network is designed to infer the adjacency relations of text components, in which the adjacency relationship between text components can be obtained independently of the spatial distance between text components.

The remainder of this paper is organized as follows. Section 2 presents an overview of previously published methods. Section 3 introduces the proposed method, and Section 4 reports the results of experiments conducted using the proposed system on the ICDAR2013, MSRA-TD500, and CTW-1500 datasets. Finally, conclusions and suggestions for future work are provided in Section 5.

## 2. Related Work

In this section, we summarize recent progress in arbitrary or irregular shape text detection in scene images. Further information about text detection in scenes is available in [1,9]. In recent years, detecting multioriented, arbitrary, or irregular text has become a popular topic in the domain of text detection in scenes, and numerous studies [5,6,8,10,11,12,13,14,15] concerning irregular text detection have been published. These methods can be roughly divided into the following three categories.

**Regression-based methods.** These methods treat the problem of scene text detection as a regression problem with text bounding boxes [14,16,17]. Textboxes [17] utilizes a sliding window and a convolution kernel for long rectangular candidate boxes to adapt to the shapes of text lines, but this method can only process horizontal texts and cannot work well when encountering arbitrarily shaped text. ABCnet [14] uses eight Bezier curve control points to describe arbitrarily shaped text, but it requires high-precision point controls, which seriously affects the final results. EAST [18] was designed to yield fast and accurate text detection results in natural scenes. Zhang et al. [16] proposed a novel adaptive boundary proposal network for arbitrary shape text detection, which can learn to directly produce accurate boundaries for arbitrary shape text without any postprocessing. Although these methods have achieved good performance in horizontal and multioriented text detection cases, they may fail to detect scene texts with large aspect ratios and directions.

**Segmentation-based methods.** These methods locate text instances based on pixel-level classification [11,12,19,20]. They usually employ deep convolutional neural networks to obtain the text segmentation region, and then several postprocessing steps are used to obtain the final text boxes. PilexLink [12] separates texts lying close to each other by predicting the pixel connections between different text instances. PSEnet [11] was proposed as a progressive scale expansion postprocessing algorithm, yielding greatly improved detection accuracy. Tian et al. [19] proposed a pixel embedding algorithm that calculates the feature distances between pixels and groups the pixels in the segmentation results according to these feature distances. These methods have obvious advantages when dealing with the segmentation of text and nontext data. However, during the training process with the text segmentation region, some irrelevant noncharacters may also be marked as characters. This may result in text line adhesion in the segmentation results of these methods, which can affect their segmentation performance.

**Connected component-based methods.** These methods usually detect individual text parts or characters first and then link or group the detected individual text parts or characters into final text instances by a postprocessing procedure. Because they have a more flexible representation and adapt well to irregularly shaped text, these methods are popular in arbitrarily shaped text detection [4,6,7,8,12,13]. PixelLink [12] separates texts lying close to each other by predicting pixel connections between different text instances. TextSnake [7] used ordered disks and text centerlines to model text instances, which made it able to represent text in arbitrary shapes. However, TextSnake still needs time-consuming and complicated postprocessing steps (centralizing, striding, and sliding) during inference. DRRG [8] also proposed a text detection method, in which each text instance is constructed by a series of ordered rectangular components(including text components and nontext components). These methods based on individual text parts generally divide the text regions into many parts (text parts and nontext parts), and many nontext parts are generated simultaneously and are fed to the following steps, which may increase the complexity and difficulty of calculation. Wei et al. [10] proposed a multioriented text detection method that generated character candidates by exhaustive segmentation. CRAFT [13] predicted character region maps and affinity maps by weakly supervised learning. As the text components contain only character regions, these methods can decrease the number of text components fed to subsequent steps and reduce the complexity and difficulty of calculations.

**Relational inference.** The connected component-based methods are usually robust for the lengths and shapes of text lines, but the performance of these methods mainly depend on the following text line grouping. PixelLink [12] used embedding features to provide instance information and generate the text area. In CRAFT [13], after predicting affinity maps by weakly supervised learning, the affinity map was used to group characters into a single instance. However, limited by the CNN receptive field, these methods cannot directly capture relations between distant component regions by local convolutional operators. To solve this problem, Zhang et al. [8] employed a graph convolutional network (GCN) to further reason and deduce the likelihood of linkages between the component and its neighbors based on local graphs. Their method achieved state-of-the-art performance on publicly available datasets. Zhang et al. [16] adopted an adaptive boundary deformation model combined with GCN and RNN to perform iterative boundary deformation to obtain a more accurate text instance shape. Their method achieved impressive results on challenging text-in-the-wild datasets such as TotalText [21].

## 3. Proposed Method

An overview of the overall architecture of our method is shown in Figure 1, and the whole framework is mainly divided into text component extraction, local inference graph establishment, deep adjacency relation inference, and text line generation. First, the text regions are extracted by the feature extraction network, and then the character text components are obtained according to the character center points in the text regions. Second, according to the basic properties of character text components, the local inference graph is established, which contains the basic connection relations between the character text components. Based on the local inference graph, the deep relational inference network further performs reasoning on the connection relationships between the character text components. Finally, the reasoning results are divided into different text instances according to the division of the connected regions.

### 3.1. Text Component Extraction

Text component extraction is an important preprocessing step whose main goal is to precisely generate text components, which are used for the subsequent text line grouping process. Different from other methods, we employ a two-stream convolutional neural network framework, which combines text region generation with character center point estimation to generate text components. This step can reduce the number of noncharacter text components without dropping the recall rate of the text component detection results.

**Text region generation.** To obtain the text region (TR), we use feature pyramid networks with VGG16 as the backbone to extract the feature maps of the given image. Finally, the feature maps are processed by classification and regression to obtain the text region. The architecture of the feature extraction network is shown in Figure 2.

**Character center point estimation.** To more accurately locate text components, we refer to the CRAFT method [13]. A fully convolutional network with VGG16_BN as the backbone is used to obtain a heatmap, which represents the probability that the pixel is the center of the character area. Then, the character center points are estimated based on the heatmaps. The workflow of character center point estimation is shown in Figure 3. First, the connected regions are obtained by threshold segmentation from the heatmap, and then the center points of the characters are obtained by the Otsu algorithm. The character center points will be directly used in the subsequent text component generation. The restriction of the character center point can ensure that the extracted text components are character text components and that the character text components are more representative in terms of feature attributes. On the other hand, the restriction of the character center point reduces the generation of nontext components, which can reduce the computational complexity.

**Text component extraction.** The subsequent task is text component generation. Different from DRRG, we combine the text region with character center point estimation to generate the text components, which can reduce the number of noncharacter text components. The processing flow of this step is shown in Figure 3. First, we divide the character center points into several groups according to the number of text regions, and the character center points in each group are sorted according to their *X* coordinate values. Second, we draw a straight line between two adjacent character center points in each group. As shown in Figure 4c, let $L_{i}$ represent the straight line that passes through $p_{i}$ and $p_{i+1}$, and let $K_{i}$ be perpendicular to the line $L_{i}$ passing through the character point $p_{i}$. Let $t_{i}$ and $b_{i}$ represent the intersection points between line $K_{i}$ and the boundaries of region $R_{1}$. Then, the associated text component is represented by the rotated rectangle whose height (denoted by $H_{i}$) is the distance from point $t_{i}$ to point $b_{i}$. The width (denoted by $W_{i}$) of the rotated rectangle is one-third of its height. According to this method, each character center point corresponds to a text component. More details are shown in Algorithm 1.
**Algorithm 1:** Text component extraction**Input:** Character center point set *P* = {$p_{1}$,$p_{2}$…$p_{n}$}; Text regions *R* = {$R_{1}$,$R_{2}$…$R_{m}$}. 
**Output:** Text components TC = {$TC_{1}$,$TC_{2}$…$TC_{n}$}
1:*S*←NULL2:**for** *i* = 1:m **do**3:      $G_{i}$←NULL4:      **for** each *p* in *P* but not in *S* **do**5:            **if**
*p* is in $R_{i}$ **then**6:                 add *p* to $G_{i}$7:                 add *p* to *S*8:**for** *i* = 1:m **do**9:      The points in $G_{i}$ are sorted by their *X* coordinate values10:**for** *i* = 1:m **do**11:      **for** each *p* in $G_{i}$ **do**12:           $L_{p}$ ←A line connecting node *p* to the next node13:           $K_{p}$ ←A line perpendicular to $L_{p}$ and passing through node *p*14:           $t_{p}$,$b_{p}$←The boundary of $R_{i}$ intersects two points of $K_{p}$15:           $H_{p}$← distance($t_{p}$, $b_{p}$)16:           $W_{p}$← $H_{p}$/317:           $\theta_{p}$← The angle between $K_{p}$ and the *X*-axis18:           $TC_{p}$← ($X_{p}$,$Y_{p}$,$H_{p}$,$W_{p}$,$\theta_{p}$)19:           add $TC_{p}$ to TC20:**return**  TC


**Detection loss.** The text component extraction loss consists of two losses and is computed as
(1)$$L=L_{cls}+L_{reg}$$
where $L_{cls}$ is a smooth *L*1 [22] regression loss and $L_{reg}$ is a cross-entropy classification loss. The classification loss $L_{cls}$ is computed as
(2)$$L_{cls}=L_{tr}+\lambda_{1}L_{tp}+\lambda_{2}L_{tn}$$
where $L_{tr}$ represents the loss for the TR; $L_{tp}$ only calculates pixels inside the TR, and $L_{tn}$ only calculates the pixels outside TR. In scene images, the text instances usually occupy a small area. Thus, the numbers of text pixels and nontext pixels are rather imbalanced. To make the network training process focus more on pixels that are hard to distinguish, we adopt the OHEM [23] strategy for the TR loss, in which the ratio between the negatives and positives is set to 3:1. In our experiments, the weights $\lambda_{1}$ and $\lambda_{2}$ are empirically set to 1.0 and 0.5, respectively. The regression loss $L_{r}$ is computed as
(3)$$L_{reg}=L_{H}+L_{\theta }$$
(4)$$L_{H}=smooth_{L1}\frac{1}{n}\sum_{i=1}^{n}\left(\tilde{H_{i}}-H_{i}\right)$$
(5)$$L_{\theta }=smooth_{L1}\frac{1}{n}\sum_{i=1}^{n}\left(\tilde{\theta_{i}}-\theta_{i}\right)$$
where $H_{i}$ and $\theta_{i}$ are ground-truth values; $\tilde{H_{i}}$ and $\tilde{\theta_{i}}$ are the corresponding predicted values; *H* is the height of the text component in the ground truth; and *n* is the number of text components.

### 3.2. Local Inference Graph Establishment

After extracting the text components, the next task is to group the text components into text lines. We treat text line grouping as a problem of inferring the adjacency relations between text components with a graph convolution network. Each node in the graph corresponds to a text component. However, if all nodes on the fully connected graph are directly used for inference, the training time and difficulty will be increased. To solve this problem, DRRG [8] uses a method that generates a local inference graph, which consists of the pivot node and its first-order and second-order neighbor nodes. The first-order neighbor nodes consist of the 8 nearest neighbor nodes of the pivot, and the second-order neighbor nodes consist of the 4 nearest neighbors of the first-order neighbor nodes. Different from DRRG, our method only needs to consider four neighbor nodes, and we select the pivot and four first-order adjacent nodes and two second-order adjacent nodes to build the local inference graph, which can reduce the number of nodes participating in the reasoning calculation. More details about the local inference graph establishment process are shown in Figure 5. In our method, adjacency is judged by the similarity between nodes, and the similarity $E_{s}$ between the pivot node *p* and another node *q* is defined as:(6)$$E_{s}=1-E_{pq}/\max\left(H,W\right)$$
(7)$$E_{pq}=\sqrt{{\left(X_{p}-X_{q}\right)}^{2}+{\left(Y_{p}-Y_{q}\right)}^{2}}$$
where *H* and *W* are the height and width of the corresponding images, respectively. $E_{pq}$ is the Euclidean distance between node *p* and node *q*.

### 3.3. Deep Adjacency Relation Inference

The local inference graph contains the basic adjacency relations between the text component nodes. However, simple link mapping or embedding mapping cannot fully reflect the adjacency relations between text component nodes, and we thus employ a deep relational inference network based on a GCN to further infer the adjacency relations between the text component nodes. In the deep adjacency relation inference process, we only need to infer the adjacency relations between the pivot and the first-order adjacent nodes, and we believe that the features of a node can also be affected by the features of its surrounding nodes, so the second-order adjacent nodes are used to provide fusion features for the first-order adjacent nodes. The inputs of the GCN usually include the feature matrix and adjacency matrix (denoted by *X* and *A*, respectively). The computational method of the two matrices is as follows:

**Feature matrix X.** The text components within the same text instance have similar geometric features, and each text component is made up of a rotated rectangle, so we combine the deep features with the geometric features as features for the text components. After extracting to a text component, we map its features to the RROI-Align layer and then obtain its deep features. Next, we obtain the geometric features of the text component through its five geometric attributes(*X*,*Y*,*W*,*H*,θ). According to the literature [8,24,25], the geometric attributes of text components are embedded in high dimensional spaces to obtain geometric features. The formulas for the embedding calculation are shown in Equations (Equation 8) and (Equation 9). Finally, we concatenate the deep features and geometric features to obtain the feature matrix *X* of the text components.
(8)$$\varepsilon_{2i}\left(z\right)=cos\left(\frac{z}{1000^{2i/C_{\varepsilon }}}\right),i\in \left(0,C_{\varepsilon }/2i-1\right)$$
(9)$$\varepsilon_{2i+1}\left(z\right)=sin\left(\frac{z}{1000^{2i/C_{\varepsilon }}}\right),i\in \left(0,C_{\varepsilon }/2i-1\right)$$

**Adjacency matrix A.** We extract the adjacency matrix *A* based on the preliminary connection relations between the text component nodes in the local inference graph. For each local inference graph, if the text component node *i* is connected with text component node *j*, let $A_{\left(i,j\right)}$ = 1, otherwise, let $A_{\left(i,j\right)}$ = 0. We do not need to explore the adjacency relation between the node and itself, and define $A_{\left(i,i\right)}$ = 0.

**Graph convolutional network.** After the feature matrix *X* and the adjacency matrix *A* of the local inference graph are obtained, we employ the deep relation inference network based on a GCN to further infer the adjacency relations between the pivot and its first-order adjacent text component nodes. $F^{k}$ represents the feature matrix of the output at layer *k*, and the convolution layer is defined as follows:(10)$$F^{k}=\sigma \left(\left(X^{k}⊕GX^{k}\right)W^{k}\right)$$
(11)$$G={\tilde{D}}^{-1/2}\tilde{A}{\tilde{D}}^{-1/2}$$
(12)$${\tilde{D}}_{i,i}=\sum_{j}{\tilde{A}}_{i,j}$$
where $X^{k}$ ∈ $R^{N\times d_{in}}$, $F^{k}$ ∈ $R^{N\times d_{out}}$, and $d_{in}$, $d_{out}$ are the feature dimensions of the input nodes and output nodes, respectively; Λ is the diagonal matrix and *N* is the number of text components in the local inference graph; *G* represents the symmetric normalized Laplacian of size *N* × *N*; the ⊕ operator denotes concatenation along the feature matrix; $W^{k}$ is the trainable weight matrix of layer *k*; and σ represents a nonlinear activation function. We only back-propagate the gradients for the nodes on the 1-order neighbors in training, because we only care about the linkage between a pivot and its first-order neighbors. For testing, we also only consider the classification of 1-order nodes.

### 3.4. Text Line Generation

After completing the deep adjacency relation inference, the adjacency probability matrix (denoted by *S*) is obtained by summarizing the probability information of all local inference graphs. Let TH represent the threshold of the adjacency probability. If the adjacency probability between node *i* and node *j* is greater than the threshold TH, the edge between them is preserved (we also say that node *i* connects node *j*) and we let ${\tilde{S}}_{\left(i,j\right)}=1$, otherwise ${\tilde{S}}_{\left(i,j\right)}=0$. Then, we use breadth-first search (BFS) to find the connected subgraphs, and the set of subgraphs is represented by *L* = {$L_{1}$,$L_{2}$…$L_{k}$}. Finally, the nodes in each subgraph are sorted, and each element in the set *L* represents a final text line that we need to find.

## 4. Experiments

### 4.1. Datasets and Evaluation Methods

**ICDAR2013 dataset.** The ICDAR2013 dataset is inherited from the benchmark used in ICDAR 2011. Several images that are duplicated over training and testing sets of the ICDAR 2011 dataset are removed. In addition, a small portion of the ground-truth annotations is revised. A total of 229 images are used for training, and 233 images are used for testing. However, this dataset is a collection of natural images having horizontal and near-horizontal text appearances.

**MSRA-TD500 dataset.** The MSRA Text Detection 500 Database (MSRA-TD500) contains 500 natural images, which are taken from indoor (office and mall) and outdoor (street) scenes using a pocket camera. The indoor images are mainly signs, doorplates, and caution signs, while the outdoor images are mostly guide boards and billboards in complex backgrounds. The resolutions of the images vary from 1296 × 864 to 1920 × 1280. The text in this dataset may be in different languages (Chinese, English, or a mixture of both), fonts, sizes, colors, and orientations.

**CTW-1500 dataset.** The SCUT-CTW1500 dataset contains 1500 images: 1000 for training and 500 for testing. In particular, it provides 10,751 cropped text instance images, including 3530 with curved text. The images are manually harvested from the internet, image libraries such as Google Open-Image, or phone cameras. The dataset contains much horizontal and multioriented text.

To evaluate the detection performance of the proposed method on the above three datasets, the evaluation method proposed by Wolf and Jolion [26] is adopted to compare the detection performance of our method with that of other methods through precision (*P*), recall (*R*), and F-measure (*F*).

### 4.2. Implementation Details

The backbone of our network is the VGG16 model pretrained on ImageNet [27]. In the training phase, the input image size is set to 640 × 640, and data augmentations (resizing, flipping, rotation, cropping, padding, etc.) are also employed to produce better-performing models. During the training process, stochastic gradient descent (SGD) is selected as the optimizer of the model, and the initial learning rate is set to 0.001. Then, the learning rate decreases by 0.0001 every 100 epochs. All experiments are carried out on a computer equipped with an RTX3080 GPU graphics card and the Linux16.04 system, and the experimental environment includes Python3.6 and PyTorch1.7.

### 4.3. Ablation Study

#### 4.3.1. Ablation Study on Text Component Extraction

To verify the effectiveness of our method in the text component generation stage, we conduct a text component ablation experiment on the MSRA-TD500 and CTW-1500 datasets. Our method is improved on the basis of DRRG, and we compare the experimental results with those of DRRG. We visualize the generated text components of the two methods, and the comparison results are shown in Figure 6. We also statistically compare the numbers of text components and detection results of the two methods. Through the data analysis comparison in Table 1, we find that the number of text components per image and the detection time decrease significantly, with the number of text components decreasing by 32% and 26%, and the detection time decreasing by 40% and 51%. Compared with those of DRRG, the F measures of our method on MSRA-TD500 and CTW-1500 are improved by 2.3% and 3.5%, respectively. The experimental results show that our text component generation method can both decrease the number of noncharacter text components and also improve the performance of text detection.

#### 4.3.2. Ablation Experiment on Deep Relation Inference

Methods that directly detect text regions through feature extraction networks often encounter difficulties when dealing with segmentation between text lines. For example, two text regions are easy to segment into one region, such as a and b in Figure 7, or one region can be divided into two regions, such as c, d, e, and f in Figure 7. The relation inference network can achieve improved text region segmentation through the adjacency relationships of the text components and then produce better detection results. To verify the effectiveness of the adjacency inference network, we also conduct deep relation inference ablation experiments on the MSRA-TD500 and CTW-1500 datasets. It can be found from Table 2 that the precision, recall, and F-measure are increased by 6.5%, 6.6%, and 6.6%, respectively, on MSRA-TD500. On CTW-1500, the precision, recall, and F-measure are increased by 3.6%, 4.8%, and 4.3%, respectively. The improvement of these performance indices also proves the effectiveness of our proposed adjacency inference network.

### 4.4. Experimental Results and Discussion

**Experiments on the ICDAR2013 dataset.** We conducted experiments on the ICDAR2013 dataset, and some examples of the experimental results are shown in Figure 8. The results show that the method in this paper achieves a good detection effect on this dataset. The comparison between the detection results of our method and other text detection methods is shown in Table 3. The data in Table 3 show that our method achieves 92.8% precision, an 87.1% recall rate, and an 89.9% F-measure in the detection performance evaluation. The recall rate and F-measure of our method are better than those of the other methods.

**Experiments on the MSRA-TD500 dataset.** To verify the effectiveness of our method on a multilanguage scene text dataset, we also carry out experiments on the MSRA-TD500 dataset. Some examples of the experimental results are shown in Figure 9, and a comparison with the results of other methods is shown in Table 4. Table 4 shows that our method also achieves good results on the MSRA-TD500 dataset, with precision, recall rate, and F-measure values of 89.7%, 85.1%, and 87.4%, respectively. Compared with other methods, our method achieves the best recall rate and F-measure value.

**Experiments on CTW-1500 dataset.** In addition, to verify the robustness of our method in terms of detecting irregular scene text, we also select the CTW-1500 irregular and multidirectional text dataset for experiments. Some examples of the experimental results are shown in Figure 10, and the performance parameter comparison with other methods is shown in Table 5. Table 5 shows that the recall rate and F-measure of our method are better than those of other methods, reaching 85.4% and 86.1%.

**Discussion.** The experimental results show that after adding constraints, our method can reduce the generation rate of noncharacter text, which in turn reduces the detection rate of nontext and improves the recall rate. In addition, the deep relational inference network further modifies the text regions obtained by the FPN, and the text detection effect is further improved. In addition, the deep relational inference network further corrects the text regions obtained by the FPN, and the text detection effect is further improved. However, our method is not ideal when dealing with the detection of overlapping text (Figure 11a), low-resolution text (Figure 11b), and partially occluded text (Figure 11c). The main reason for this is that the character detection effect is impacted during the detection process.

### 4.5. Generalization Ability

To further verify the effectiveness of our method, we randomly collected some images of natural scenes containing irregular text from the internet and conducted experiments on the collected images. The experimental results are shown in Figure 12. As shown in Figure 12, the proposed method is robust to irregular scene texts, which shows that the proposed method has a good generalization ability. We believe that two main factors lead to the good results of our approach. On the one hand, we employ the connected component-based method, which is robust to the shapes and sizes of scene texts, to generate text commponents in text component stage. On the other hand, the GCN can obtain a better adjacency relation between two text component nodes during the text line grouping stage. The above shows that the generalization ability of our method is good.

## 5. Conclusions

This paper presents an irregular text detection method based on a graph convolution network. In the text component extraction stage, we combine character center point estimation with the text regions to extract more accurate text components. In the text line grouping stage, we treat text line grouping as a problem of inferring the adjacency relations of text components. From the experimental results, it is shown that our method is effective and robust in irregular scene text detection scenarios and achieves promising results on three public datasets. Future work will focus on two aspects.

(1) We will solve the limitations of the current methods, such as the detection of overlapping text, low-resolution text, and partially occluded text in scene text images.

(2) Combining our method with text recognition, we will design an end-to-end text detection and recognition approach.

## Figures and Tables

**Figure 1 sensors-23-01070-f001:**
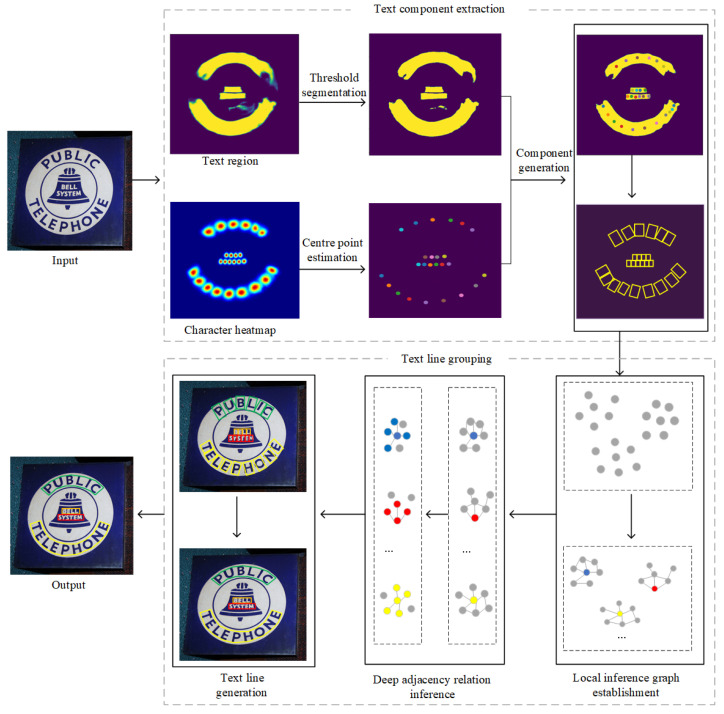
Overview of our overall architecture. Our network mainly consists of four parts: Text component extraction, local inference graph establishment, deep adjacency relation inference, and text line generation.

**Figure 2 sensors-23-01070-f002:**
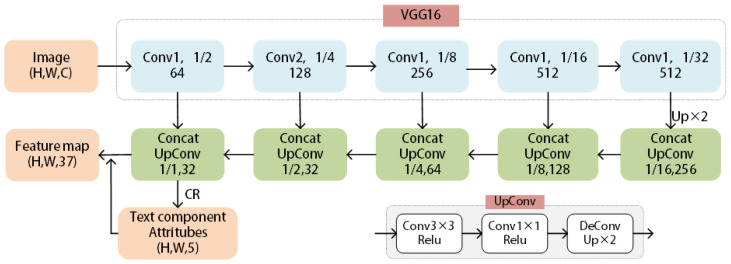
Architecture of the feature extraction network, where CR represents the classification and regression operation for the text components. $CR\in R^{H\times W\times 5}$ has 2 channels for the logistic classification of text regions and 3 channels for the logistic regression of h1, h2, and θ.

**Figure 3 sensors-23-01070-f003:**
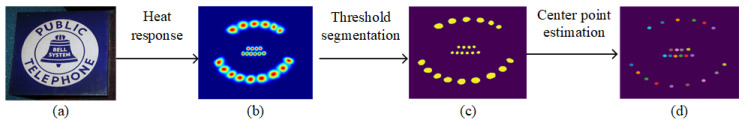
The workflow of character center node estimation. (**a**) Source image. (**b**) Heatmap. (**c**) Segmentation result. (**d**) Character center points.

**Figure 4 sensors-23-01070-f004:**
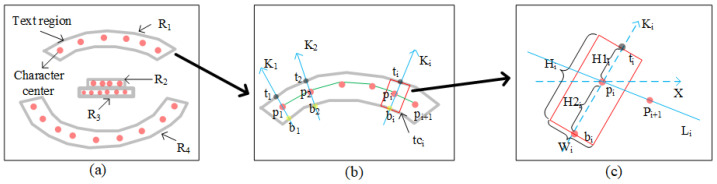
The processing flow of text component generation. (**a**) The result of character center point grouping. (**b**) Text region $R_{1}$ and its character center points. (**c**) One of the text components.

**Figure 5 sensors-23-01070-f005:**
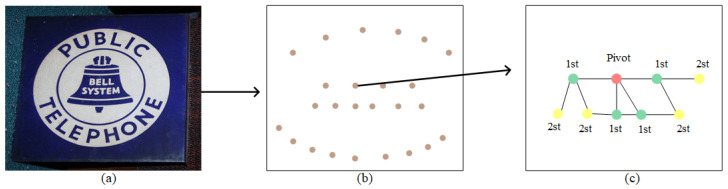
Local inference graph generation. (**a**) Source image. (**b**) Graph of text component nodes. (**c**) Local inference graph. “1st” indicates the first-order nodes and “2st” denotes the second-order nodes.

**Figure 6 sensors-23-01070-f006:**
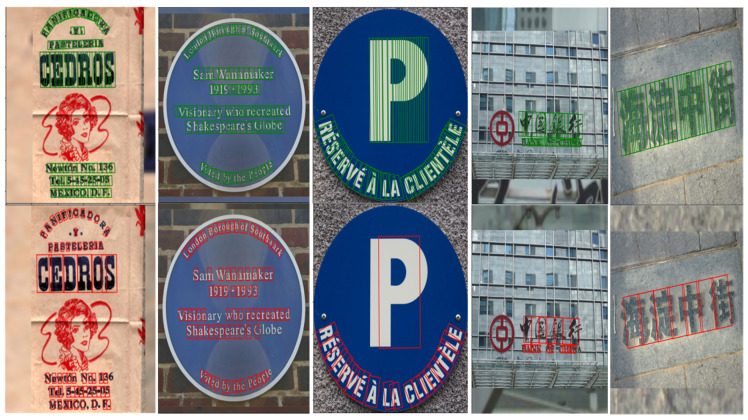
Comparative examples of the text component extraction results of our method and DRRG. The first row contains the text component instances extracted by our method, and the second row contains text component instances extracted by DRRG.

**Figure 7 sensors-23-01070-f007:**
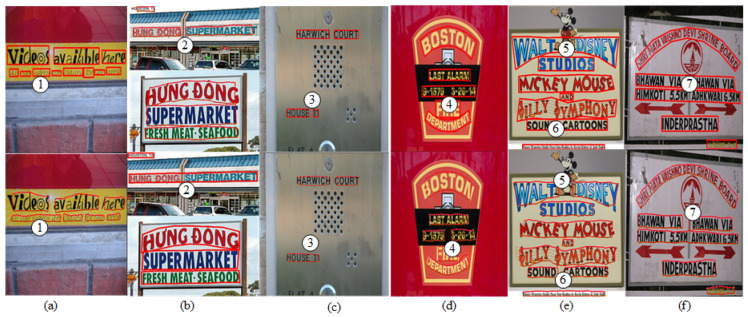
Experimental results of two detection methods. The images in the top row are the detection results obtained directly through the feature extraction network, and the images in the next row are the detection results obtained with the addition of the GCN-based relation inference network.

**Figure 8 sensors-23-01070-f008:**
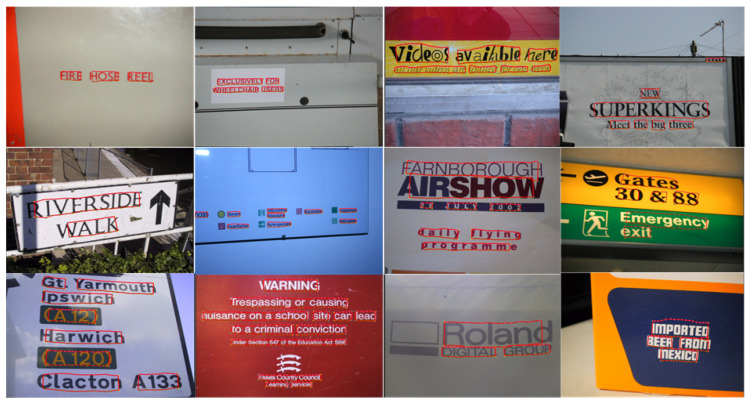
Detection examples of the proposed method in the ICDAR2013 dataset.

**Figure 9 sensors-23-01070-f009:**
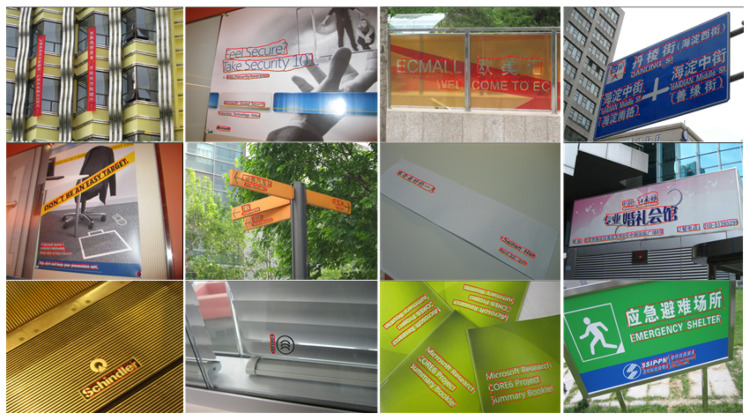
Detection examples obtained by the proposed method on the MSRA-TD500 dataset.

**Figure 10 sensors-23-01070-f010:**
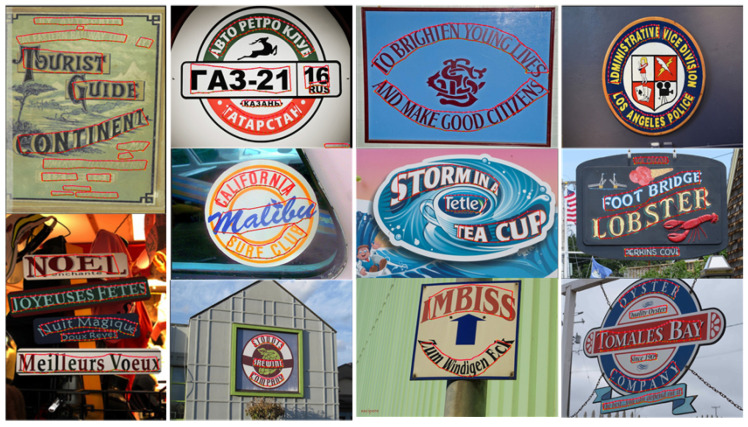
Detection examples produced by the proposed method on the MSRA-TD500 dataset.

**Figure 11 sensors-23-01070-f011:**
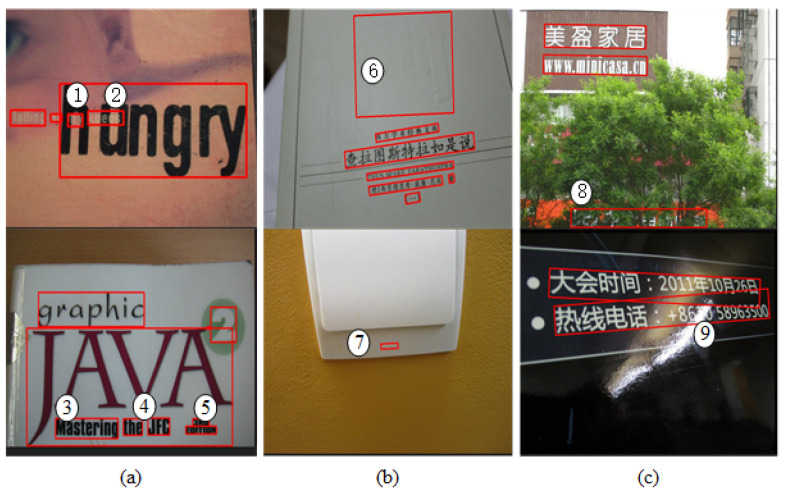
Some failed detection examples of the proposed method: (**a**) overlapping text; (**b**) low-resolution text; (**c**) partially occluded text. The red lines mark text boxes, and the text markers ①–⑨ represent the text that is un-detected.

**Figure 12 sensors-23-01070-f012:**
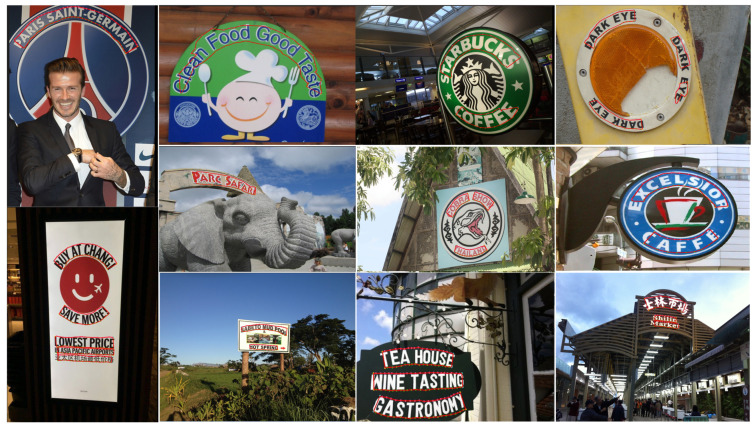
Examples of internet and life scene text images.

**Table 1 sensors-23-01070-t001:** Results of ablation experiments involving text component extraction.

Datasets	Methods	Ave_num	Ave_time(s)	P(%)	R(%)	F(%)
MSRA-TD500	DRRG	41	0.252	88.1	82.3	85.1
	Ours	28	0.151	89.7	85.1	87.4
CTW-1500	DRRG	84	0.374	83.8	81.5	82.6
	Ours	62	0.183	86.7	85.4	86.1

**Note:** Ave_num is the average number of text components per image; Ave_time is the average detection time per image.

**Table 2 sensors-23-01070-t002:** Results of ablation experiments of deep relation inference.

Datasets	Methods	P (%)	R (%)	F (%)
MSRA-TD500	Baseline	83.2	78.5	80.8
	Baseline+GCN	89.7	85.1	87.4
CTW-1500	Baseline	83.1	80.6	81.8
	Baseline+GCN	86.7	85.4	86.1

**Note:** Baseline is directly obtained by text region; Baseline+GCN is a relation inference network based on GCN added to Baseline.

**Table 3 sensors-23-01070-t003:** Experimental results of various methods on the ICDAR2013 dataset.

Methods	P (%)	R (%)	F (%)
Wei Y et al. [28]	83.5	77.2	80.2
Wei Y et al. [10]	87.3	81.1	84.3
Gao et al. [29]	90.0	80.0	85.0
CAST [30]	**94.0**	69.0	80.0
Ours	92.8	**87.1**	**89.9**

**Table 4 sensors-23-01070-t004:** Experimental results of various methods on the MSRA-TD500 dataset.

Methods	P (%)	R (%)	F (%)
SegLink [31]	86.0	70.0	77.0
TextField [32]	87.4	75.9	81.3
CRAFT [13]	88.2	78.2	82.9
Wan et al. [33]	81.6	77.2	79.3
Wang et al. [34]	85.0	82.0	83.0
DRRG [8]	88.1	82.3	85.1
R-YOLO [35]	90.2	81.9	85.8
Zobeir Raisi et al. [36]	90.9	83.8	87.2
Liao et al. [37]	**91.5**	83.3	87.2
Ours	89.7	**85.1**	**87.4**

**Table 5 sensors-23-01070-t005:** Experimental results of various methods on the CTW-1500 dataset.

Methods	P (%)	R (%)	F (%)
TextDragon [38]	84.5	82.8	83.6
TextField [32]	83.0	79.8	81.4
Zhang et al [16]	**87.7**	80.6	84.0
CRAFT [13]	86.0	81.1	83.5
Wan et al. [33]	85.1	78.2	81.5
DRRG [8]	85.9	83.0	84.4
CountourNet [39]	83.7	84.1	83.9
PCENet [5]	87.6	83.4	85.5
Ours	86.7	**85.4**	**86.1**

## Data Availability

The data used to support the findings of this study are available from the corresponding author upon request.

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
