# Peer review of "Irregular Scene Text Detection Based on a Graph Convolutional Network"

_sensors, 2023, doi:10.3390/s23031070_

Round 1
Reviewer 1 Report (New Reviewer)
In general, the proposed scheme is interesting and the simulation results seem to be promising. Unfortunately, the proposed schemes are not clearly described and it is difficult for the reviewer to verify the correctness and to evaluate the value of the proposed method. So :
The novelty of the paper should be more clearly emphasized. A more comprehensive review and discussion of methods investigating text detection based on a graph convolutional network may be included.
Author Response
Dear reviewer:
Thank you for the prompt handling of our paper (sensors-2151596), entitled "Irregular scene text detection based on a graph convolutional network" by Shiyu Zhang, Caiying Zhou, Yonggang Li, Xianchao Zhang, Lihua Ye, Yuanwang Wei to Sensors.
We would like to express our sincere thanks to you for the valuable comments, which helped with revising the paper to make it sound and clear. We have extensively revised the manuscript accordingly. The major improvements include the following:
1) We have corrected and polished the language and presentation in the revised manuscript through cooperation with a professional editing service.
2) We have revised and supplemented some contents in the “abstract” and “introduction”.
3) To refine the statement of the “Related work” section, we rewrote this section in the revised manuscript and supplemented a more comprehensive review and discussion of methods investigating text detection based on a graph convolutional network in the last paragraph of the “Related work” section.
4) We have modified Figure 1 and updated it in the revised manuscript.
5) We have supplemented some content about the Character centre point estimation and updated it in the revised manuscript.
6) Minor modifications have been made to Figure 3 and Figure 4.
7) More detailed descriptions have been supplemented in the revised manuscript.
We provide the following response to explain how each comment is addressed in the revised manuscript. If you need any other information, please do not hesitate to contact me. Responses to the comments, Please see the attachment

Reviewer 2 Report (New Reviewer)
The paper proposes an approach that can detect irregular text in natural scene images, which first employ a fully convolutional network architecture based on VGG16_BN to generate text components via the estimated character center points, and then text line grouping is treated as a problem of inferring the adjacency relations of text components with a graph convolution network (GCN). The article is sound and complete. The proposed method achieves promising results on three public datasets.
Very small notes for readability:
-- The text in the Figure 1 should be clearer;
Author Response
Dear reviewer:
Thank you for the prompt handling of our paper (sensors-2151596), entitled "Irregular scene text detection based on a graph convolutional network" by Shiyu Zhang, Caiying Zhou, Yonggang Li, Xianchao Zhang, Lihua Ye, Yuanwang Wei to Sensors.
We would like to express our sincere thanks to you for the valuable comments, which helped with revising the paper to make it sound and clear. We have extensively revised the manuscript accordingly. The major improvements include the following:
1) We have corrected and polished the language and presentation in the revised manuscript through cooperation with a professional editing service.
2) We have revised and supplemented some contents in the “abstract” and “introduction”.
3) To refine the statement of the “Related work” section, we rewrote this section in the revised manuscript and supplemented a more comprehensive review and discussion of methods investigating text detection based on a graph convolutional network in the last paragraph of the “Related work” section.
4) We have modified Figure 1 and updated it in the revised manuscript.
5) We have supplemented some content about the Character centre point estimation and updated it in the revised manuscript.
6) Minor modifications have been made to Figure 3 and Figure 4.
7) More detailed descriptions have been supplemented in the revised manuscript.
We provide the following response to explain how each comment is addressed in the revised manuscript. If you need any other information, please do not hesitate to contact me. Responses to the comments, Please see the attachment

Reviewer 3 Report (New Reviewer)
The paper presents a novel method to detect irregular text in natural scene images. In fact, the authors employ a fully convolutional network architecture based on VGG16_BN to generate text components via the estimated character center points. Moreover, text line grouping is treated as a problem of inferring the adjacency relations of text components with a graph convolution network. The authors used three public datasets to evaluate the proposed method which are ICDAR2013, CTW-1500 and MSRA-TD500. The obtained results show promising performance gains.
The paper is well written however many shortcomings must be treated to ameliorate the manuscript both in content and form.
1) The writing of the paper should be improved.
2) Figure 1 should be more detailed.
3) The paper would be significantly improved with the addition of more details about the Character center point estimation.
Author Response
Dear reviewer:
Thank you for the prompt handling of our paper (sensors-2151596), entitled "Irregular scene text detection based on a graph convolutional network" by Shiyu Zhang, Caiying Zhou, Yonggang Li, Xianchao Zhang, Lihua Ye, Yuanwang Wei to Sensors.
We would like to express our sincere thanks to you for the valuable comments, which helped with revising the paper to make it sound and clear. We have extensively revised the manuscript accordingly. The major improvements include the following:
1) We have corrected and polished the language and presentation in the revised manuscript through cooperation with a professional editing service.
2) We have revised and supplemented some contents in the “abstract” and “introduction”.
3) To refine the statement of the “Related work” section, we rewrote this section in the revised manuscript and supplemented a more comprehensive review and discussion of methods investigating text detection based on a graph convolutional network in the last paragraph of the “Related work” section.
4) We have modified Figure 1 and updated it in the revised manuscript.
5) We have supplemented some content about the Character centre point estimation and updated it in the revised manuscript.
6) Minor modifications have been made to Figure 3 and Figure 4.
7) More detailed descriptions have been supplemented in the revised manuscript.
We provide the following response to explain how each comment is addressed in the revised manuscript. If you need any other information, please do not hesitate to contact me. Responses to the comments, Please see the attachment

Round 2
Reviewer 1 Report (New Reviewer)
The paper is now more fair and ready to be published.
This manuscript is a resubmission of an earlier submission. The following is a list of the peer review reports and author responses from that submission.
Round 1
Reviewer 1 Report
1. Detecting irregular text in natural scene images is a challenging task that has recently attracted considerable attention from research communities. This task is also one of the typical applications of artificial intelligence and machine learning algorithms. Therefore, many algorithms based on bionics are worth quoting, such as Deep Neural Network Regression for Automated Retinal Layer Segmentation in Optical Coherence Tomography Images, Floating pollutant image target extraction algorithm based on immune extremum region. The author should explain the practicability of the paper in more detail.
2. Can the author analyze which features are used to detect irregular text in natural scene images in the paper's methods. Can these characteristics be obtained by other methods? If other methods are available, why is this method better than others?
3. Can the author give a mathematical proof or principle description to describe the generalization performance of the algorithm?
4. The logic of the paper is clear, and the author's scientific research attitude is very serious.
Author Response
Response to Reviewer 1 Comments
Point 1: Detecting irregular text in natural scene images is a challenging task that has recently attracted considerable attention from research communities. This task is also one of the typical applications of artificial intelligence and machine learning algorithms. Therefore, many algorithms based on bionics are worth quoting, such as Deep Neural Network Regression for Automated Retinal Layer Segmentation in Optical Coherence Tomography Images, Floating pollutant image target extraction algorithm based on immune extremum region. The author should explain the practicability of the paper in more detail.
Response 1:
Thank you for your suggestions. We have carefully reviewed the relevant literature and updated to the paper. According to your suggestions, we have explained the application scenarios and practicability of irregular scene text detection, more detail information can be found in Section 1.
Point 2: Can the author analyze which features are used to detect irregular text in natural scene images in the paper's methods? Can these characteristics be obtained by other methods? If other methods are available, why is this method better than others?
Response 2:
Our text component features are mainly composed of two parts, which are depth features and geometric features. The deep features are obtained by the FPN network with VGG16 as the backbone. The geometric features are obtained by mapping the five metrics (x, y, w, h, r) of the text components.
Most of the previous methods only extract deep features, which are usually extracted by neural networks such as VGG19 and resnet50. The depth feature only contains color and shape information of the text component, and ignores the position and length and width attribute information of the text component. Compared with these methods, our method synthesizes the depth feature and geometric feature to represent the text components, and the fused features are more representative. Different from DRRG, the geometric features are extracted from the character components in our method, which can dramatically reduce the number of non-character text components and ensure a high text component detection recall rate at the same time.
In order to fully explain the advantages of our method and DRRG in detection efficiency, we updated the ablation study of text component extraction, more details can be found in subsection 4.3.1 in the paper.
Point 3: Can the author give a mathematical proof or principle description to describe the generalization performance of the algorithm?
Response 3:
Our method is based on the connected component, which is not affected by the length, direction and shape of scene text, and has strong generalization ability.
In order to further illustrate the generalization performance of our algorithm, we found some irregular scene text images on the Internet for detection, and some experimental results are shown as follow.
(The figure is attached in the attachement)
Point 4: The logic of the paper is clear, and the author's scientific research attitude is very serious.
Response 4:
Thank you for your recognition of our work.
Point 5:
Moderate English changes required.
Response 5:
We are very sorry for our incorrect writing which disturbs the readability. We read the whole paper carefully and corrected the grammatical errors and written errors.

Reviewer 2 Report
The authors present an interesting approach for detecting irregular areas containing text. The manuscript is well organized and presents sufficient, relevant, and recent references. The authors describe with sufficient detail their approach and make clear their contribution. They compare their approach with several other algorithms and show promising results.
It would be interesting to show time performance as well as the possible challenges to taking this approach to embedded systems.
Author Response
Point 1: It would be interesting to show time performance as well as the possible challenges to taking this approach to embedded systems.
Response 1:
Thank you for your suggestions. We will try to migrate our methods to embedded platforms in our future work.

Round 2
Reviewer 1 Report
This paper is not innovative enough, and it is recommended to revise it carefully.